# Barter Exchange with Shared Item Valuations

## ABSTRACT

In barter exchanges agents enter seeking to swap their items for other items on their wishlist. We consider a centralized barter exchange with a set of agents and items where each item has a positive value. The goal is to compute a (re)allocation of items maximizing the agents' collective utility subject to each agent's total received value being comparable to their total given value. Many such centralized barter exchanges exist and serve crucial roles; e.g., kidney exchange programs, which are often formulated as variants of directed cycle packing. We show finding a reallocation where each agent's total given and total received values are equal is NP-hard. On the other hand, we develop a randomized algorithm that achieves optimal utility in expectation and where, i) for any agent, with probability 1 their received value is at least their given value minus $v^*$ where $v^*$ is said agent's most valuable owned and wished-for item, and ii) each agent's given and received values are equal in expectation. Our algorithm builds on the dependent rounding techniques from Gandhi et al. [16].

## KEYWORDS

Barter-Exchanges, Centralized exchanges, Community Markets

**ACM Reference Format:**
Anonymous Author(s). 2023. Barter Exchange with Shared Item Valuations. In *Proceedings of ACM Conference (Conference'17)*. ACM, New York, NY, USA, 13 pages. https://doi.org/10.1145/nnnnnnn.nnnnnnn

## 1 INTRODUCTION

Social media platforms have recently emerged into small scale business websites. For example, platforms like Facebook, Instagram, etc. allow its users to buy and sell goods via verified business accounts. With the proliferation of such community marketplaces, there are growing communities for buying, selling and exchanging (swapping) goods amongst its users. We consider applications, viewed as Barter Exchanges, which allow users to exchange board games, digital goods, or any physical items amongst themselves. For instance, the subreddit GameSwap[1] (61,000 members) and Facebook group BoardgameExchange[2] (51,000 members) are communities where users enter with a list of owned video games and board games. The existence of this community is testament to the fact that although users could simply liquidate their goods and subsequently purchase

[1] www.reddit.com/r/Gameswap
[2] https://www.facebook.com/groups/boardgameexchange

the desired goods, it is often preferable to directly swap for desired items. Additionally, some online video games have fleshed out economies allowing for the trade of in-game items between players while selling items for real-world money is explicitly illegal e.g., Runescape[3]. In these applications, a centralized exchange would achieve greater utility, in collective exchanged value and convenience, as well as overcome legality obstacles.

A centralized barter exchange market provides a platform where agents can exchange items directly, without money/payments. Beyond the aforementioned applications, there exist a myriad of other markets facilitating the exchange of a wide variety of items, including books, children's items, cryptocurrency, and human organs such as kidneys. There are both centralized and decentralized exchange markets for various items. HomeExchange[4] and ReadItSwapIt[5] are decentralized marketplaces that facilitate pairwise exchanges by mutual agreement of vacation homes and books, respectively. Atomic cross chain swaps allow users to exchange currencies within or across various cryptocurrencies [e.g., 18, 24]. Kidney exchange markets [see, e.g., 2, 4] and children's items markets (e.g., Swap[6]) are examples of centralized exchanges facilitating swaps amongst incompatible patient-donor pairs and children items or services amongst parents. Finding optimal allocations is often NP-hard. As a result heuristic solutions have been explored extensively [17, 23].

Currently, the aforementioned communities GameSwap and BoardGameExchange make swaps in a decentralized manner between *pairs* of agents, but finding such pairwise swaps is often inefficient and ineffective due to demanding a "double coincidence of wants" [19]. However, centralized *multi-agent* exchanges can help overcome such challenges by allowing each user to give and receive items from possibly different users. Moreover, the user's goal is to swap a subset a subset of their owned games for a subset of their desired games with comparable (or greater) value. Although an item's value is subjective, a natural proxy is its re-sale price, which is easily obtained from marketplaces such as Ebay.

We consider a centralized exchange problem where each agent has a have-list and a wishlist of distinct (indivisible) items (e.g., physical games) and, more generally, each item has a value agreed upon by the participating agents (e.g., members of the GameSwap community). The goal is to find an allocation/exchange that (i) maximizes the collective utility of the allocation such that (ii) the total value of each agent's items before and after the exchange is equal.[7] We call this problem *barter with shared valuations*, BarterSV, and it is our subject of study. Notice that bipartite perfect matching is a special case of BarterSV where each agent has a single item in both its have-list and wishlist each and where all the items are have the same value. On the other hand, we show BarterSV is NP-Hard (Theorem 2).

[3] www.jagex.com/en-GB/terms/rules-of-runescape
[4] www.homeexchange.com
[5] www.readitswapit.co.uk
[6] www.swap.com
[7] Equivalently, the total value of the items given is equal to the total value of the items received.

In the following sections we formulate BarterSV as bipartite graph-matching problem with additional *barter constraints*. Our algorithm BarterDR is based on rounding the fractional allocation of the natural LP relaxation to get a feasible integral allocation. A direct application of existing rounding algorithms (like [15]) to BarterSV results in a worst-case where agents give away all their items and receive none in exchange. This is wholly unacceptable for any deployed centralized exchange. In contrast, our main result ensures BarterDR allocations have reasonable net value for all agents; more precisely each agent gives and receives the same value in expectation and the absolute difference between given and received values is at most the value of their most valuable item (Theorem 1).

## 1.1 Problem formulation: BarterSV

Suppose we are given a ground set of items $\mathcal{I}$ to be swapped, item values $\{v_j \in \mathbb{R}^+ : j \in \mathcal{I}\}$ where $\mathbb{R}^+$ denotes the non-negative real numbers, and a community of agents $i \in [n]$ where $[n]$ denotes $\{1, 2, \ldots, n\}$. Each agent $i$ possesses items $H_i \subseteq \mathcal{I}$ and has wishlist $W_i \subseteq \mathcal{I}$. Further, each agent $i$ also has a "capacity" function $\eta_i : H_i \to \mathbb{N}$ denoting the number of copies agent $i$ has of each item; similarly, $\omega_i : W_i \to \mathbb{N}$ denotes a cap on the number of copies agent $i$ desires of each item. We allow agents to swap an arbitrary number of copies of the same item as a natural generalization of the original problem.

A *valid allocation* of these items involves agents swapping their items with other agents that desire said item while ensuring no agent neither gives more copies of an item than they own nor receives more copies than desired. The goal of BarterSV is to find a valid allocation of maximum utility *subject to* no agent giving away more value than they received. The following lemmas in this section greatly simplify the problem's presentation; their full justification is deferred to the appendix. Let $\eta = \max_{i \in [n], j \in H_i} \eta_i(j)$ and similarly $\omega = \max_{i \in [n], j \in H_i} \omega_i(j)$.

LEMMA 1. *Any instance of* BarterSV *with arbitrary item capacity functions* $\eta_i$ *and* $\omega_i$, *for each agent* $i \in [n]$, *can be reduced to a corresponding* BarterSV *instance with unit capacities (i.e., for all valid* $i$ *and* $j$, $\eta_i(j), \omega_i(j) = 1$) *in time* $\mathrm{poly}(|\mathcal{I}|, n, \log \eta, \log \omega)$.

**Definition 1.** *Value-balanced Matching (VBM)* Suppose there is bipartite graph $G = (L, R, E)$ with vertex values $v_a > 0\ \forall a \in L \cup R$. Let each edge $e \in E$ have weight $w_e \in \mathbb{R}$, $L = \dot{\bigcup}_i L_i$, and $R = \dot{\bigcup}_i R_i$. For a given matching $M \subseteq E$, let $V_i^{(L)} = \sum_{\ell:(\ell,r) \in M, \ell \in L_i} v_\ell$ and $V_i^{(R)} = \sum_{r:(\ell,r) \in M, r \in R_i} v_r$, where $\dot{\bigcup}$ denotes disjoint union. The goal of VBM is to find $M$ of maximum weight subject to, for each $i$, the value of items matched in $L_i$ and $R_i$ are equal i.e., $V_i^{(R)} = V_i^{(L)}$.

LEMMA 2. BarterSV *is equivalent to VBM.*

Given a BarterSV instance we reduce it to a corresponding instance BarterSV with unit capacities (as Lemma 1 would suggest), and then reduce once more to a corresponding VBM instance via the construction of an appropriate bipartite graph and valuation function. Thus VBM is the technical lens through which we view BarterSV in the remainder of the paper. The construction is as follows. For each agent $i \in [n]$, build the vertex sets $L_i := \{\ell_{ij} : j \in H_i\}$ and $R_i := \{r_{ij} : j \in W_i\}$. Then the bipartite graph of interest has

vertex sets $U = \dot{\bigcup}_{i \in [n]} L_i$ and $V = \dot{\bigcup}_{i \in [n]} R_i$, as well as edge set $E := \{(\ell_{ij}, r_{i'j}) : j \in H_i \cap W_{i'}\}$, Thus, for each item $j \in \mathcal{I}$, we draw an edge between all left and right vertices corresponding to $j$. Each vertex $\ell_{ij}$ and $r_{ij}$ has value $v_j$; so $v(\ell_{ij}) = v_j$ and so on. Crucially, $E$ has edges only between vertices of equal value. Then a valid allocation corresponds to a VBM $M$ in $(U, V, E)$ such that for each $e \in E$, $y_e = 1$ if $e \in M$ and $y_e = 0$ otherwise is feasible in the following Integer Program (IP);

$$\max \quad \sum_{e \in E} w_e y_e \tag{1a}$$

$$\text{subj. to} \quad y(\ell_{ij}) \leq 1, \qquad i \in [n], \ell_{ij} \in L_i \tag{1b}$$

$$y(r_{ij}) \leq 1, \qquad i \in [n], r_{ij} \in R_i \tag{1c}$$

$$\sum_{a \in L_i} y(a)v_a = \sum_{b \in R_i} y(b)v_b, \qquad i \in [n] \tag{1d}$$

$$y_e \in \{0, 1\}, \qquad e = (\ell_{ij}, r_{i'j}) \in E. \tag{1e}$$

The weights $w_e \in \mathbb{R}$ can be set arbitrarily; this detail will be elaborated upon shortly. For $a \in U \cup V$, we denote $y(a) := \sum_{e \in N(a)} y_e$ where $N(a)$ denotes the open neighborhood of $a$ i.e., $N(a) := \{(a, b) \in E : b \in U \cup V\}$, and $\mathbb{Z}^+$ denotes the non-negative integers. Thus $e = (\ell_{ij}, r_{i'j}) \in M$ says agent $i$ gives item $j$ to agent $i'$. With this in mind, (1b) ensures each agent $i$ gives item $j$ away at most once, (1c) ensures each agent $i$ receives at most one copy of item $j$, and (1d) ensures, for each agent $i$, the value received $\sum_{b \in R_i} y(b)v_b$ equals the value given $\sum_{a \in L_i} y(a)v_a$ (i.e., $M$ is a VBM with $V_i^{(R)} = V_i^{(R)}$). It follows that an allocation is a valid allocation if and only if the corresponding $\{y_e\}$ is a feasible point of (1); i.e., Lemma 2. For each $e \in E$ we may set $w_e = v_j$ and recover the objective of maximizing the collective value received by all agents. Nevertheless, our results hold even if $w_e$ is set arbitrarily. For example, the algorithm designer could place greater value on certain item allocations, or they may maximize the sheer number of items received by uniformly setting $w_e = 1$. Henceforth $\sum_{e \in E} w_e y_e$ is the *allocation's utility*. By relaxing (1e) to $y_e \geq 0$ for $e \in E$ we arrive at the natural LP relaxation of BarterSV, namely BarterSV-LP. The following lemma means BarterSV guarantees will follow from carefully rounding the related BarterSV-LP solution.

LEMMA 3. *IP* (1) *is equivalent to* BarterSV. *Moreover, the objective of* BarterSV-LP *is an upper bound on the objective of IP* (1).

## 2 PRELIMINARIES: GKPS DEPENDENT ROUNDING

Our results build on the dependent rounding algorithm due to [15], henceforth referred to as GKPS-DR. GKPS-DR is an algorithm that takes $\{x_e\} \in [0, 1]^{|E|}$ defined over the edge set $E$ of a biparite graph $(L, R, E)$ and outputs $\{X_e\} \in \{0, 1\}^{|E|}$. In each iteration GKPS-DR considers the graph of floating edges (those edges $e$ with $0 < x_e < 1$) and selects a maximal path or cycle $P \subseteq E$ on floating edges. The edges of $P$ are decomposed into alternate matchings $M_1$ and $M_2$ and rounded in the following way. Fix $\alpha^{\mathrm{GKPS}} = \min \{\gamma > 0 : (\bigvee_{e \in M_1} x_e + \gamma = 1) \vee (\bigvee_{e \in M_2} x_e - \gamma = 0)\}$, and $\beta^{\mathrm{GKPS}} = \min \{\gamma > 0 : (\bigvee_{e \in M_2} x_e + \gamma = 1) \vee (\bigvee_{e \in M_1} x_e - \gamma = 0)\}$. Thus, each $x_e$ is updated to $x'_e$ according to one of the following

disjoint events: with probability $\frac{\beta^{\text{GKPS}}}{\alpha^{\text{GKPS}}+\beta^{\text{GKPS}}}$

$$x'_e = \begin{cases} x_e + \alpha, & e \in M_1 \\ x_e - \alpha, & e \in M_2 \end{cases}; \quad \text{else,} \quad x'_e = \begin{cases} x_e - \beta, & e \in M_1 \\ x_e + \beta, & e \in M_2. \end{cases}$$

The selection of $\alpha$ and $\beta$ ensures at least one edge is rounded to 0 or 1 in every iteration. GKPS-DR guarantees **(P1)** *marginal*, **(P2)** *degree preservation*, and **(P3)** *negative correlation* properties:

**(P1)** $\forall e \in E, \Pr(X_e = 1) = x_e$.

**(P2)** $\forall a \in L \cup R$ and with probability 1, $X(a) \in \{\lfloor x(a) \rfloor, \lceil x(a) \rceil\}$.

**(P3)** $\forall a \in L \cup R, \ \forall S \subseteq N(a), \ \forall c \in \{0,1\}, \Pr\left(\bigwedge_{s \in S} X_s = c\right) \leq \prod_{s \in S} \Pr(X_s = c)$.

*Remark 1.* When GKPS-DR rounds a path between vertices $a$ and $b$, the signs of the changes to $x(a)$ and $x(b)$ are equal if and only if $a$ and $b$ belong to different graph sides.

## 3 RELATED WORK

Centralized barter exchanges have been studied by several others in the context of kidney-exchanges [2–4]. BarterSV generalizes a well-studied kidney-exchange problem in the following way. The Kidney Exchange Problem (KEP) is often formulated as directed cycle packing in compatibility patient-donor graphs [2] where each node in the graph corresponds to a patient-donor pair and directed edges between nodes indicate compatibility. Abraham et al. [2], Biró et al. [5] observed this problem reduces to bipartite perfect matching, which is solvable in polynomial-time. We show BarterSV is NP-Hard and thus resort to providing a randomized algorithm with approximate guarantees on the agents' net values via LP relaxation followed by *dependent rounding*.

There has been extensive work on developing dependent rounding techniques, that round the fractional solution in some correlated way to satisfy both the hard constraints and ensure some negative dependence amongst rounded variables that can result in concentration inequalities. For instance, the hard constraints might arise from an underlying combinatorial object such as a packing [6], spanning tree [8], or matching [15] that needs to be produced. In our case, the rounded variables must satisfy both matching (1b), (1c), and barter constraints (1d) (i.e., each agent gives the items of same total value as it received). Gandhi et al. [15] developed a rounding scheme where the rounded variables satisfy the matching constraints along with other useful properties. Therefore, we adapt their rounding scheme (to satisfy matching constraints) followed by a careful rounding scheme that results in rounded variables satisfying the barter constraints.

Centralized barter exchanges are well-studied under various barter settings. For instance, Abraham et al. [2] showed that the bounded length *edge-weighted directed cycle packing* is NP-Hard which led to several heuristic based methods to solve these hard problems, e.g., by using techniques of operations research [7, 9, 17, 23], AI/ML modeling [21, 22]. Recently several works focused on the fairness in barter exchange problems [1, 13, 14, 20]. Our work adds to the growing body of research in theory and heuristics surrounding ubiquitous barter exchange markets.

## 4 OUTLINE OF OUR CONTRIBUTIONS AND THE PAPER

Firstly, we introduce the BarterSV problem, a natural generalization of *edge-weighted directed cycle packing* and show that it is NP-Hard to solve the problem exactly. Our main contribution is a randomized dependent rounding algorithm BarterDR with provable guarantees on the quality of the *allocation*. The following definitions help present our results. Suppose we are given an integral allocation $\{X_e\} \in \{0,1\}^{|E|}$, we define the *net value loss* of each agent $i$ (i.e., the violation in the barter constraint (1d)):

$$D_i := \sum_{b \in L_i} v_b X(b) - \sum_{a \in R_i} v_a X(a). \tag{2}$$

Our main contribution is a rounding algorithm BarterDR that satisfies both *matching* (1b), (1c) and *barter* constraints (1d) as desired in *multi-agent* exchanges. Recollect that existing rounding algorithm such as GKPS-DR (indeed a pre-processing step of our BarterDR) rounds the fractional matching to an integral solution enjoying the properties mentioned in Section 2. The main challenge in our problem is satisfying the *barter constraint*. Here, a direct application of GKPS-DR alone can result in a worst case violation of $\sum_{a \in L_i} v_a$ on $D_i$, corresponding to the agent losing all their items and gaining none (see the example in the Appendix). However, our algorithm BarterDR rounds much more carefully to ensure, for each agent $i$, $D_i$ is at most $v_i^* := \max_{a \in L_i \cup R_i} v_a$, i.e., the most valuable item in $H_i \cup W_i$. The two following theorems provide lower and upper bounds on tractable $D_i$ (i.e., (2)) guarantees for BarterSV. BarterDR on a bipartite graph $(L, R, E)$ is worst-case time $O((|L| + |R|)(|L| + |R| + |E|))$ where $L, R = O(|\mathcal{I}|n)$. We view Theorem 1 as our main result.

THEOREM 1. *Given a* BarterSV *instance,* BarterDR *is an efficient randomized algorithm achieving an allocation with optimal utility in expectation and where, for all agents $i$, $D_i < v_i^*$ with probability 1 and $\mathbb{E}[D_i] = 0$.*

THEOREM 2. *Deciding whether a* BarterSV *instance has a non-empty valid allocation with $D_i = 0$ for all agents $i$ is NP-hard, even if all item values are integers.*

Owing to its similarities to GKPS-DR, BarterDR enjoys similar useful properties:

THEOREM 3. BarterDR *rounds $\{x_e\} \in [0,1]^{|E|}$ in the feasible region of* BarterSV-LP *into $\{X_e\} \in \{0,1\}^{|E|}$ while satisfying **(P1)**, **(P2)**, and **(P3)**.*

*Outline of the paper.* In Section 5 we describe BarterDR (Algorithm 1), our randomized algorithm for BarterSV, and its subroutines FindCCC and CCWalk in detail. Next, we give proofs and proof sketches for Theorems 1 and 3.

## 5 BARTERDR: DEPENDENT ROUNDING ALGORITHM FOR BarterSV

*Notation.* BarterDR is a rounding algorithm that proceeds in multiple iterations, therefore we use a superscript $r$ to denote the value of a variable at the beginning of iteration $r$. An edge $e \in E$ is said to be floating if $x_e^r \in (0,1)$. Analogously, let $E^r := \{e \in E : x_e^r \in (0,1)\}$, a vertex $a \in L \cup R$ is said to be floating if $x^r(a) := \sum_{e \in E} x_e^r \notin$

$\mathbb{Z}$ and the sets of floating vertices are $L^r := \{a \in L : x^r(a) \notin \mathbb{Z}\}$ and $R^r := \{a \in R : x^r(a) \notin \mathbb{Z}\}$. Define $C(i) := \{a : a \in L_i \cup R_i\}$, for each $i \in [n]$, to be the set of participating vertices in each barter constraint. We say two vertices $a, b \in L \cup R$ are partners if there exists $i \in [n]$ such that $a, b \in C(i)$ and $a \neq b$. Note if $a$ and $b$ are partners, then they are distinct vertices corresponding to items (owned or desired) by the same agent $i$. In iteration $r$, a vertex $a \in C(i)$ is said to be *partnerless* if $C(i) \cap (L^r \cup R^r) = \{a\}$; i.e., $a$ is the only floating vertex in $C(i)$. We use the shorthand $a \sim b$ to denote $a$ and $b$ are partners. Edges and vertices not floating are said to be *settled*. For vertices $a$ and $b$, $a \rightsquigarrow b$ denotes a simple path from $a$ to $b$. Define $D_i^r$ to be $D_i$, as defined in (2), but with variables $\{x_e^r\}$ instead of $\{X_e\}$. The fractional degree of $a \in L \cup R$ refers to $x^r(a)$.

Once an edge is settled, its value does not change. In the each iteration BARTERDR looks exclusively at the floating edges $E^r$ and the graph induced by them. Namely, $G^r := (L(E^r), R(E^r), E^r)$ where $L(E^r) := \{a \in L : \exists e \in E^r, \ e \in N(a)\}$ and $R(E^r)$ is defined analogously. In each iteration, at least one edge or vertex becomes settled, i.e., $|E^r| + |L^r| + |R^r| > |E^{r+1}| + |L^{r+1}| + |R^{r+1}|$. Therefore BARTERDR terminates in iteration $T$ where $|E^T| = 0$ and $T \leq |L| + |R| + |E|$.

*Algorithm and analysis outline.* BARTERDR begins by making $G$ acyclic via the pre-processing step in Section 5.1. Next, BARTERDR proceeds as follows. While there are floating edges find an appropriate sequence of paths $\mathcal{P}$ constituting a CCC or CCW (defined in Section 5.2). The strategy for judiciously rounding $\mathcal{P}$ is fleshed out in Section 5.3. Finally, Section 5.4 concludes with proof sketches for Theorems 1 and 3.

## 5.1 Pre-processing: remove cycles in $G$

The pre-processing step consists of finding a cycle $C$ via depth-first search in the graph of floating edges and rounding $C$ via GKPS-DR until there are no more cycles. Let $\{x_e^0\}_{e \in E}$ denote the LP solution and $\{x_e^1\}_{e \in E}$ denote the output of the pre-processing step. BARTERDR begins on $\{x_e^1\}_{e \in E}$.

GKPS-DR on cycles never changes fractional degrees, i.e., $\forall a \in L \cup R, \ x^0(a) = x^1(a)$. Lemma 5 is used to construct CCC's and CCW's, and it is the raison d'être for the pre-processing step.

LEMMA 4. *The pre-processing step is efficient and gives $D_i^1 = 0$ for all agents $i$ with probability 1.*

LEMMA 5. *Each connected component of $G^1$ has at least 2 floating vertices.*

## 5.2 Construction of CCC's and CCW's via FINDCCC

This section introduces CCC's and CCW's. The definition of these structures facilitates rounding edges while respecting the barter constraints each iteration. The subroutines for constructing CCC's and CCW's, FINDCCC and CCWALK, are described in Algorithms 2 and 3. The correctness of these subroutines, and thus the existence of CCC's and CCW's, follows from Lemma 6.

**Definition 2.** A *connected component cycle* (CCC) is a sequence of $q \geq 1$ paths $\mathcal{P} = \langle s_1 \rightsquigarrow t_1, \dots, s_q \rightsquigarrow t_q \rangle$ such that, letting $V(\mathcal{P}) = \bigcup_{i \in [q]} \{s_i, t_i\}$ be the paths' endpoint vertices,

(1) $\forall i \in [q], t_i \sim s_{i+1}$ (taking $s_{q+1} \equiv s_1$),
(2) $\forall a \in V(\mathcal{P}), |V(\mathcal{P}) \cap C(a)| = 2$,
(3) $\forall i \in [q], s_i \rightsquigarrow t_i$ belong to distinct connected components, and
(4) $\forall i \in [q], s_i$ and $t_i$ are floating vertices.

Instead, we have a *connected component walk* (CCW) if criteria 3) and 4) are met but 1) and 2) are relaxed to: 1) $\forall i \in [q-1], t_i \sim s_{i+1}$ and $s_1$ and $t_q$ are partnerless; and 2) $\forall a \in V(\mathcal{P}) - \{s_1, t_q\}, |V(\mathcal{P}) \cap C(a)| = 2$.

Recall that a rounding iteration $r$ is fixed so whether a vertex is floating or partnerless is well-defined. When $\mathcal{P}$ is rounded the set of vertices whose fractional degrees change is precisely $V(\mathcal{P})$. Requirements 1 and 2 of a CCC say $t_i$ and $s_{i+1}$ are partners *and* they do not have any other partner vertices in $V(\mathcal{P})$. Comparably, for CCW's these requirements imply the same for all vertices but the "first" and "last," which are partnerless. Therefore, for CCC's and CCW's the vertices in $V(\mathcal{P})$ respectively appear in $q$ and $q + 1$ distinct barter constraints. The requirements in the definitions of CCC and CCW come in handy during the analysis because: each path belongs to a different connected component hence they are vertex and edge disjoint; if a barter constraint has exactly two vertices in $V(\mathcal{P})$ then these vertices' fractional degree changes can be made to cancel each other out in the barter constraint; and floating vertices ensure paths can be rounded in a manner analogous to GKPS-DR. For comparison, GKPS-DR also needed paths with floating endpoints, but maximal paths always have such endpoints whereas the paths of $\mathcal{P}$ need not be maximal. Consequently, the requirement that paths of $\mathcal{P}$ have floating endpoints must be imposed.

---

**Algorithm 1:** BARTERDR

**Input:** $\{x_e^1\} \in \{0, 1\}^{|E|}$, corresponding to
$G^1 = (L(E^1), R(E^1), E^1)$; i.e., the output of the
pre-processing described in Section 5.1

1  $r \leftarrow 1$
2  **while** $E^r \neq \emptyset$ **do**
3      $\mathcal{P} \leftarrow$ CCC or CCW returned by FINDCCC in $G^r$
4      Round $\mathcal{P}$ as described in Section 5.3 yielding $G^{r+1}$ and
     $\{x_e^{r+1}\}$;    $r \leftarrow r + 1$
5  **end**
6  **return** $\{x_e^r\} \in \{0, 1\}^{|E|}$

---

*Uncrossing the half-CCWs.* We show how to resolve "crossing" half-CCW's as mentioned in FINDCCC Line 9. Using $O_1$ and $O_2$ build $\bar{V} := \langle s_q', t_{q-1}', \dots, s_2', t_1', t_1, s_2, \dots, s_q \rangle$. $\bar{V}$ can be seen as the sequence of path endpoints (i.e., $V$ in CCWALK) resulting from a run of CCWALK$(s_q')$. By the half-CCW's "crossing" we mean that in some iteration of the while-loop of CCWALK either a connected component is revisited or $t_i$ was partners with a vertex previously visited. But these cases are precisely Lines 9 and 14 from CCWALK where it is known a CCC can be resolved.

LEMMA 6. *If $G^r$ has no cycles, FINDCCC efficiently returns a CCC or CCW.*

---

**Algorithm 2:** FindCCC

**Input:** $G^r = (U(E^r), V(E^r), E^r)$

**Output:** CCC or CCW $\langle s_i \rightsquigarrow t_i \rangle_{i \in [q]}$

1   $s_1, t_1 \leftarrow$ distinct floating vertices in some connected component $C$ of $G_r$

2   $(O_1, \sigma_1) \leftarrow$ CCWalk$(t_1)$

3   **if** $\sigma_1 = $ "CCC" **then**

4     |   **return** $O_1$

5   $(O_2, \sigma_2) \leftarrow$ CCWalk$(s_1)$

6   **if** $\sigma_2 = $ "CCC" **then**

7     |   **return** $O_2$

8   Let $O_1 = (t_1, s_2, t_2, \ldots, s_q, t_q)$ and $O_2 = (t'_1, s'_2, t'_2, \ldots, s'_{q'}, t'_{q'})$

9   If $O_1$ and $O_2$ "cross," resolve $O_1$ and $O_2$ into a CCC and return it; see Section 5.2

10   **return**

$$\left\langle t'_{q'} \rightsquigarrow s'_{q'}, t'_{q'-1}, s'_{q'-1}, \ldots, s'_2 \rightsquigarrow t'_2, t'_1 \rightsquigarrow t_1, s_2 \rightsquigarrow t_2, \ldots, s_q \rightsquigarrow t_q \right\rangle$$

---

**Algorithm 3:** CCWalk$(a)$

**Input:** $G^r = (U(E^r), V(E^r), E^r)$, the walk's starting vertex $a$

**Output:** A CCC or half of a CCW; a string indicating whether a CCC was returned

1   $V \leftarrow (t_1)$, letting $t_1 := a$     // ordered list of path endpoints

2   $S \leftarrow \{C_1\}$ where $C_1$ is the connected component containing $a$     // seen CC's

3   $i \leftarrow 2$

4   **while** *True* **do**

5     |   **if** $t_{i-1}$ *is partnerless* **then**

6     |     |   **return** $(V, "CCW")$

7     |   $s_i \leftarrow$ partner of $t_{i-1}$

8     |   $C_i \leftarrow$ connected component containing $s_i$

9     |   **if** $C_i \in S$ **then**

10     |     |   $C_i = C_k$ for some $k < i$ so let $s'_k := s_i$

11     |     |   **return** $(\langle s'_k \rightsquigarrow t_k, \ldots, s_{i-1} \rightsquigarrow t_{i-1} \rangle, "CCC")$

12     |   $t_i \leftarrow$ floating vertex in $C_i$ distinct from $s_i$

13     |   $V \leftarrow V \oplus (s_i)$, where $\oplus$ denotes sequence concatenation

14     |   **if** $\exists b \in V, b \sim t_i$ **then**     // $t_i$ already has a partner in $V$

15     |     |   It must be that $b$ already had a partner $c \in V$. WLOG, $b = t_{k-1}$ and $c = s_k$, some $k \le i$

16     |     |   **return** $(\langle s_k \rightsquigarrow t_k, \ldots, s_i \rightsquigarrow t_i \rangle, "CCC")$

17     |   $V \leftarrow V \oplus (t_i)$

18     |   $i \leftarrow i + 1, \quad S \leftarrow S \cup \{C_i\}$

19   **end**

---

## 5.3   Rounding CCC's and CCW's

Now we shed light on what we mean by carefully rounding the paths of the CCC/CCW $\mathcal{P}$. But first we build some intuition. Focus on $t_p$ and $s_{p+1}$ for some fixed $1 \le p \le q$ in case of a CCC (or $p < q$ in the case of a CCW). Since $t_p \sim s_{p+1}$, whatever rounding procedure we use, we want the relative signs of the changes to $x^r(t_p)$ and $x^r(s_{p+1})$ to depend on whether $t_p$ and $s_{p+1}$ fall on the same or different sides of $G$ (these sides being "left" and "right" corresponding to vertex sets $L$ and $R$; equivalently, left and right of "=" in (1d)). This way (1d) is preserved after rounding. Likewise, the magnitudes of fractional degree changes to $t_p$ and $s_{p+1}$ must be balanced depending on $v_p$ and $v_{p+1}$ so that (1d) is preserved for $i$ corresponding to $t_p, s_{p+1} \in C(i)$. Intuitively, these two points are necessary to successfully round $\mathcal{P}$. To this end, we now define roundable colorings. If $a$ and $b$ are vertices belonging to different sides of the graph, we say $a \perp b$; otherwise we say $a \not\perp b$.

**Definition 3** (Roundable coloring). The CCC $\mathcal{P} = \langle s_i \rightsquigarrow t_i \rangle_{i \in [q]}$ has a *roundable coloring* if there exists $C : V(\mathcal{P}) \to \{-1, 1\}$ such that i) for all $i \in [q]$, $C(s_i) = C(t_i)$ if and only if $s_i \perp t_i$; and ii) for all $i \in [q]$, $C(t_i) = C(s_{i+1})$ if and only if $t_i \perp s_{i+1}$. A roundable coloring for a CCW is defined the same way except ii) becomes $\forall i \in [q-1], C(t_i) = C(s_{i+1})$ if and only if $t_i \perp s_{i+1}$.

LEMMA 7. *Every CCC and CCW admits an efficiently computable roundable coloring $C$.*

Property i) will ensure $a, b \in V(\mathcal{P})$ see same-sign fractional degree change if and only if $C(a) = C(b)$. Property ii) is equivalent to Remark 1 and verifies each $s_i \rightsquigarrow t_i$ is roundable in GKPS-DR manner.

Although the notation used next is cumbersome, the intuition is to fix $\alpha, \beta > 0$ "small enough" that all edge variables stay in $[0, 1]$ and vertex fractional degrees stay within their current ceilings and floors but "large enough" that at least one edge or vertex is settled. First, fix the roundable coloring $C$, which is possible per Lemma 7. Next, decompose each path $s_i \rightsquigarrow t_i$ into alternating matchings $M^i_{-1}$ and $M^i_1$ such that $\forall a \in \{s_i, t_i\}, \exists e \in M^i_{C(a)}$ such that $e \in N(a)$; property ii) of $C$ guarantees this is possible. In other words, vertex $a \in \{s_i, t_i\}$ is present in $M^i_{C(a)}$. For readability drop the $r$ superscripts briefly and let

$$\begin{aligned}
\Gamma^i_{-1}(\gamma) \equiv &\bigvee_{e \in M^i_{-1}} (x_e + \gamma = 1) \vee \bigvee_{e \in M^i_1} (x_e - \gamma = 0) \\
&\vee \bigvee_{a \in \{s_i, t_i\}} (C(a) = -1 \implies x(a) + \gamma = \lceil x(a) \rceil) \\
&\vee \bigvee_{a \in \{s_i, t_i\}} (C(a) = 1 \implies x(a) - \gamma = \lfloor x(a) \rfloor),
\end{aligned} \tag{3}$$

and, symmetrically,

$$\begin{aligned}
\Gamma^i_1(\gamma) \equiv &\bigvee_{e \in M^i_1} (x_e + \gamma = 1) \vee \bigvee_{e \in M^i_{-1}} (x_e - \gamma = 0) \\
&\vee \bigvee_{a \in \{s_i, t_i\}} (C(a) = 1 \implies x(a) + \gamma = \lceil x(a) \rceil) \\
&\vee \bigvee_{a \in \{s_i, t_i\}} (C(a) = -1 \implies x(a) - \gamma = \lfloor x(a) \rfloor).
\end{aligned} \tag{4}$$

Finally, the magnitudes fixed (in analogy to Section 2) are

$$\alpha := \min\left\{ \gamma > 0 : \bigvee_{i \in [q]} \Gamma^i_{-1}\left(\frac{1}{v_i}\gamma\right) \right\}, \ \beta := \min\left\{ \gamma > 0 : \bigvee_{i \in [q]} \Gamma^i_1\left(\frac{1}{v_i}\gamma\right) \right\}. \tag{5}$$

Both $\alpha$ and $\beta$ are well defined as they are the minima of non-empty finite sets. The update proceeds probabilistically as follows: $\forall i \in [q], \forall e \in s_i \rightsquigarrow t_i$,

$$\text{w.p. } \frac{\beta}{\alpha + \beta}, \quad x_e^{r+1} = \begin{cases} x_e^r + \frac{1}{v_i}\alpha, & e \in M_{-1}^i \\ x_e^r - \frac{1}{v_i}\alpha, & e \in M_1^i \end{cases}; \tag{6}$$

$$\text{else, w.p. } \frac{\alpha}{\alpha + \beta}, \quad x_e^{r+1} = \begin{cases} x_e^r - \frac{1}{v_i}\beta, & e \in M_{-1}^i \\ x_e^r + \frac{1}{v_i}\beta, & e \in M_1^i \end{cases}. \tag{7}$$

### 5.4 Algorithm analysis

*Proof sketch of Theorem 3.* The proof proceeds via the following invariants maintained at each iteration $r$ of BARTERDR. Except for **(J2)**, the proofs for the invariants are almost identical to those in [15]. This is because BARTERDR is crafted so as to be similar to GKPS-DR in the ways necessary for this analysis to carry over.

**(J1)** $\forall e \in E, \mathbb{E}[x_e^r] = x_e^0$.

**(J2)** $\forall a \in L \cup R$ and with probability 1, $\lfloor x^0(a) \rfloor \leq x^r(a) \leq \lceil x^0(a) \rceil$.

**(J3)** $\forall a \in L \cup R, \ \forall S \subseteq N(a), \ \forall c \in \{0,1\}, \mathbb{E}[\prod_{e \in S} x_e^{r+1}] \leq \mathbb{E}[\prod_{e \in S} x_e^r]$.

Though BARTERDR chooses $\alpha$ and $\beta$ differently, the main difference is there may not be a rounded edge in *every* path of the CCC/CCW, which is okay.

**Lemma 8.** *BARTERDR achieves optimal objective in expectation and $\forall i \in [n], \ \mathbb{E}[D_i] = 0$.*

**Lemma 9.** *If $D_i^r = 0$ and there exists distinct floating $a, b \in C(i)$, then $D_i^{r+1} = 0$.*

**PROOF OF THEOREM 1.** It is straightforward to check that after solving BarterSV-LP-Caps there are at most $|E|$ floating edges. Each iteration of the pre-processing step finds a cycle, say using depth-first-search, and rounds said cycle in time $O(|L| + |R|)$ with at least one edge being settled every time a cycle is rounded. Therefore, the pre-processing step takes time at most $O(|E| \cdot (|L| + |R|))$. Similarly, FINDCCC takes time $O(|L| + |R|)$ to find and round a CCC or CCW. Each iteration a CCC/CCW is rounded either one edge or vertex becomes settled. Therefore BARTERDR runs in time $O((|L| + |R|) \cdot (|L| + |R| + |E|))$.

Let $D_i^r$ be $D_i$ like in (2) but with variables $x_e^r$ instead of $X_e$. Then Lemma 9 guarantees that for each agent $i$, $D_i^r = 0$ implies $D_i^{r+1} = 0$ until $L_i \cup R_i$ has exactly one floating vertex (if this happens at all). This means if in some iteration the number of floating vertices in $L_i \cup R_i$ went from at least 2 to 0, then $D_i = 0$ by the degree preservation invariant **(J2)**, proved in the proof of Theorem 3, and we are done. Therefore, the only case we must consider is when there is a solitary floating vertex $d \in L_i \cup R_i$. Let $t'$ be the first iteration that started with $L_i \cup R_i$ having a sole vertex $d$ with $x^{t'}(d) \notin$

$\mathbb{Z}$. Then by expanding

$$D_i \leq \left| \sum_{a \in L_i} v_a X(a) - \sum_{b \in R_i} v_b X(b) \right| \tag{8}$$

$$= \left| \sum_{a \in L_i} v_a X(a) - \sum_{a \in L_i} v_a x^{t'}(a) + \sum_{b \in R_i} v_b x^{t'}(b) - \sum_{b \in R_i} v_b X(b) \right| \tag{9}$$

$$= \left| \sum_{a \in L_i} v_a (X(a) - x^{t'}(a)) + \sum_{b \in R_i} v_b(x^{t'}(b) - X(b)) \right| \tag{10}$$

$$\leq \sum_{a \in (L_i \cup R_i) - \{d\}} v_a \left| X(a) - x^{t'}(a) \right| + v_d \left| x^{t'}(d) - X(d) \right| \tag{11}$$

$$< v_d \leq v_i^*, \tag{12}$$

which is our desired $D_i$ bound for Theorem 1. Equation (9) follows because we assume $D_i^1 = 0$ (as $D_i^0$ corresponds to the LP solution and the pre-processing step thus guarantees $D_i^1 = 0$) and $t'$ is the first iteration where $L_i \cup R_i$ contains exactly one floating vertex; therefore, by induction and Lemma 9, $D_i^{t'} = 0$. Inequality (11) follows from the triangle inequality. The strict inequality in (12) follows because $d$ was the sole floating vertex of $L_i \cup R_i$ in iteration $t'$; hence by Lemma E.2, $\forall a \in (L_i \cup R_i) - \{d\}$, $X(a) - x^{t'}(a) = 0$ and $|x^{t'}(d) - X(d)| < 1$.

By assumption, $D_i^0 = 0$ for all $i$ since $\{x_e^0\}_{e \in E}$ is an optimal solution to the corresponding BarterSV-LP and the pre-processing step ensures $D_i^0 = 0 \implies D_i^1 = 0$. Then, by Lemma 9, the only $D_i$'s that are not necessarily preserved are those where $L_i \cup R_i$ ends up with exactly one floating vertex in some algorithm iteration $t'$. As argued above, this case leads to $D_i < v_i^*$. Together with Lemma 8 this completes the proof. □

Consequently from Theorems 1 and 3:

**COROLLARY 1.** BarterSV-LP *with all items having equal values is integral.*

## 6 FAIRNESS

Fairness is an important consideration when resource allocation algorithms are deployed in the real-world. Theorem 3 allows for adding *fairness* constraints to BarterSV-LP. Previous works such as [10–12] studied various group fairness notions, and formulated the fair variants of problems like Clustering, Set Packing, etc., by adding *fairness* constraints to the Linear Programs of the respective optimization problems.

Consider a toy example of such an approach where we are given $\ell$ communities $G_1, \ldots, G_\ell \subseteq [n]$ of agents coming together to thicken the market. In order to incentivize said communities to join the centralized exchange, the algorithm designer may promise that each community $G_p$ will receive at least $\mu_p$ units of value *on average*. By adding the constraints

$$\sum_{i \in G_p} x(r_{ij})v_j \geq \mu_p, \quad p \in [\ell] \tag{13}$$

to the BarterSV-LP, the algorithm designer ensures that the expected utility of each group $G_p$ is at least $\mu_p$. More precisely, **(P1)**

and the linearity of expectation ensures

$$\mathbb{E}\left[\sum_{i \in G_p} X(r_{ij})v_j\right] = \sum_{i \in G_p} \mathbb{E}[X(r_{ij})]v_j$$

$$= \sum_{i \in G_p} x(r_{ij})v_j \geq \mu_p.$$

The same rationale can be extended to provide individual guarantees (in expectation) by adding analogous constraints for each agent. We conclude this brief discussion by highlighting the versatility of LPs and subsequently of BarterDR.

## 7 CONCLUSION

We introduce and study BarterSV, a centralized barter exchange problem where each item has a value agreed upon by the participating agents. The goal is to find an allocation/exchange that (i) maximizes the collective utility of the allocation such that (ii) the total value of each agent's items before and after the exchange is equal. Though it is NP-hard to solve BarterSV exactly, we can efficiently compute allocations with optimal expected utility where each agent's *net value loss* is at most a single item's value. Our problem is motivated by the proliferation of large scale web markets on social media websites with 50,000-60,000 active users eager to swap items with one another. We formulate and study this novel problem with several real-world exchanges of video games, board games, digital goods and more. These exchanges have large communities, but their decentralized nature leaves much to be desired in terms of efficiency. Future directions of this work include accounting for arbitrary item valuations i.e., different agents may value items differently.

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

## A DIRECT APPLICATION OF GKPS-DR FAILS

Example 1 is a worst case instance where a direct application of GKPS-DR to the fractional optimal solution of BarterSV results in a net loss of $\sum_{j \in L_i} v_j$ for some agent $i$; i.e., agent $i$ gives away all their items and does not receive any item from its wishlist.

*Example* 1. Consider an instance of BarterSV with two agents where $W_1 = \{1, 2\}, W_2 = \{3, 4\}$ and $H_1 = \{3, 4\}, H_2 = \{1, 2\}$. Let the values of the items be $v(1) = v(2) = 10$ and $v(3) = v(4) = 20$.

Figure 1 shows the bipartite graph of the BarterSV instance from Example 1. The edges are unweighted and the optimal LP solution is $x = [0.5, 0.5, 1, 1]$. The first two coordinates of $x$ correspond to items given by agent 1. GKPS-DR will round both of these coordinates to 0 with positive probability thus resulting in agent 2 incurring a net loss of $\sum_{a \in L_2} v_a = 20$ units of value.

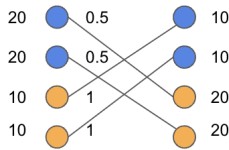

**Figure 1: The bipartite graph corresponding to the** BarterSV **instance in Example 1. Blue and orange vertices correspond to agents** 1 **and** 2**, respectively. The optimal LP solution is** $x = [0.5, 0.5, 1, 1]$**.**

*Observation* 1. GKPS-DR rounding $x$ results in the vector $X = [0, 0, 1, 1]$ with positive probability; this is the worst case for agent 2 where their net loss is $\sum_{a \in L_2} v_a = 20$ units of value.

The above example can be easily generalized to instances with larger item-lists and more agents where some agent $i$ achieves net value loss $\sum_{a \in L_i} v_a$ with positive probability.

## B REDUCING TO BarterSV WITH SINGLE ITEM COPIES

BarterSV with arbitrary item capacities can be modeled with the following integer program. Parallel edges (values of $y_e > 1$) model the fact that $i$ may give $i'$ up to $\min\{\eta_i(j), \omega_{i'}(j)\}$ copies of $j$. Each vertex $\ell_{ij}$ and $r_{ij}$ has value $v_j$; so $v_{\ell_{ij}} = v_j$ and so on. Crucially, $E$ has edges only between vertices of equal value. Let $\mathbb{Z}^+$ denote the non-negative integers. Then a valid allocation corresponds to a vector $y$ such that for $e \in E$, $y_e \in \mathbb{Z}^+$ is feasible in the following Integer Program (IP);

$$\max \quad \sum_{e \in E} w_e y_e \tag{14a}$$

$$\text{subj. to} \quad y(\ell_{ij}) \le \eta_i(j), \qquad i \in [n], \ell_{ij} \in L_i \tag{14b}$$

$$y(r_{ij}) \le \omega_i(j), \qquad i \in [n], r_{ij} \in R_i \tag{14c}$$

$$\sum_{a \in L_i} y(a)v_a = \sum_{b \in R_i} y(b)v_b, \qquad i \in [n] \tag{14d}$$

$$y_e \in \mathbb{Z}^+, \qquad e = (\ell_{ij}, r_{i'j}) \in E. \tag{14e}$$

The weights $w_e \in \mathbb{R}$ can be set arbitrarily; we will come back to this. For $a \in L \cup R$, we denote $y(a) := \sum_{e \in N(a)} y_e$ where $N(a)$ denotes the open neighborhood of $a$ i.e., $N(a) := \{(a, b) \in E : b \in U \cup V\}$. Thus for $e = (\ell_{ij}, r_{i'j}) \in E$, $y_e = k$ says agent $i$ gives $k$ copies of $j$ to agent $i'$. With this in mind, (1b) ensures each agent $i$ gives away at most $\eta_i(j)$ copies of item $j$, (1c) ensures each agent $i$ receives at most $\omega_i(j)$ copies of item $j$, and (1d) ensures, for each agent $i$, the value received $\sum_{b \in R_i} y(b)v_b$ equals the value given $\sum_{a \in L_i} y(a)v_a$. It follows that an allocation is valid allocation if and only if the corresponding $\{y_e\}_{e \in E}$ is a feasible point of (1). For each $e \in E$ we may set $w_e = v_j$ and recover the objective of maximizing the collective value received by all agents. Nevertheless, our results hold even if $w_e$ is set arbitrarily. For example, the algorithm designer could place greater value on certain item transactions, or they may maximize the sheer number of items received by uniformly setting $w_e = 1$. Henceforth $\sum_{e \in E} w_e y_e$ is the *allocation's utility*. Therefore, IP (1) is equivalent to BarterSV.

By relaxing (1e) to $y_e \ge 0$ for $e \in E$ we arrive at the natural LP relaxation of BarterSV, namely BarterSV-LP-Caps. This can be reduced to an instance of BarterSV-LP (that is, with unit capacities) as follows.

PROOF OF LEMMA 1. The instance with unit copies of each item proceeds as follows. Note that this instance will be large but it is only a thought experiment; we do not directly solve the corresponding LP, write the full graph down, etc.. Fix an agent $i$ and an item $j$ in $H_i$. Make $\eta_i(j)$ copies of this vertex each with unit capacity, say $\ell_{ij1}, \ell_{ij2}, \ldots, \ell_{ij\eta_i(j)}$. Similarly, for an item $j' \in W_i$, make $\omega_i(j')$ copies $r_{ij'1}, r_{ij'2}, \ldots, r_{ij'\omega_i(j')}$. Like before add edges between all vertices corresponding to the same items. Keep all edge weights the same and use the same corresponding weights for edges between copies i.e., if $e = (\ell_{ijk_1}, r_{i'jk_2})$ and $f = (\ell_{ij}, \ell_{i'j})$ then $w_e = w_f$. Call this new set of edges over vertex copies $E'$.

To see the two formulations are equivalent, we show $y, \forall e \in E, y_e \ge 0$ is feasible to BarterSV-LP-Caps if and only if $z, \forall e \in E', z_e \in [0, 1]$ is feasible to BarterSV-LP. Moreover, $y$ and $z$ have the same objective value. Let $e = (\ell_{ij}, r_{i'j})$ then $y_e = k + r$ for $k \in \mathbb{Z}^+$ and $0 \le r < 1$. Correspondingly let $e_p = (\ell_{ijp}, r_{i'jp}) \in E'$ and set $z_{e_1}, z_{e_2}, \ldots, z_{e_k}$ all equal to 1 and $z_{e_{k+1}} = r$. $k + r \le \min\{\eta_i(j), \omega_i(j)\}$ if and only if $y$ and $z$ are feasible. Moreover, both $y_e$ and $(z_{e_1}, \ldots, z_{e_{k+1}})$ each contribute $(k + r)w_e$ to the objective and $(k + r)v_j$ value given by agent $i$ and received by agent $i'$.

Therefore we always *write and solve* BarterSV-LP-Caps and use BarterSV-LP only as a *thought experiment* to facilitate the presentation of the problem. It is easy to check that the size of BarterSV-LP-Caps is polynomial in $|\mathcal{I}|$, $n$, $\log \eta$, and $\log \omega$. □

Following from the proof above we also have (where $E'$ corresponds to the graph with vertex copies as outlined in the proof of Lemma 1)

LEMMA B.1. *A solution $\{x_e\}_{e \in E'}$ to* BarterSV-LP *has at most $|E|$ floating variables.*

PROOF OF LEMMA B.1. Like in the proof of Lemma 1, corresponding to each group of $z_{e_1}, z_{e_2}, \ldots$ there is at most one $z_{e_p} = r$ for $0 \le r < 1$. Therefore, the number of floating edges is at most $|E|$. □

PROOF OF LEMMA 2. The reduction in the proof of Lemma 1 is an instance of VBM shown to be equivalent to BarterSV. □

PROOF OF LEMMA 3. The IP is equivalent as argued above. Since the feasible region BarterSV-LP is larger than that of the corresponding IP, it follows the objective of BarterSV-LP is an upper bound on the objective of (1). □

## C PRE-PROCESSING

PROOF OF LEMMA 9. When rounding a cycle each vertex appearing in the cycle has two edges with changes equal-in-magnitude and opposite-in-sign. Therefore, all vertex fractional degrees, and thus $D_i$ values, are preserved. □

LEMMA C.1. *Let $\{x_e\} \in (0,1)^{|E|}$ be a vector of floating edges over a connected bipartite graph $(U, V, E)$. Then the number of floating vertices in $(U, V, E)$ is not 1.*

PROOF OF LEMMA C.1. Let $u$ be the sole floating vertex, i.e., $x(u) \notin \mathbb{Z}$, and, without loss of generality, let $u \in U$. For $S \subseteq U \cup V$, let $d(S) := \sum_{s \in S} x_s$. Then $d(U - \{u\}) + x_u = d(U) = \sum_{e \in E} x_e = d(V)$. But $u$ being the only floating vertex implies $d(U - \{u\}) + x(u) \notin \mathbb{Z}$ and $d(V) \in \mathbb{Z}$. □

PROOF OF LEMMA 5. If there are 0 floating vertices in a connected component, then each vertex has degree at least two because $\forall e \in E^r, x_e^r \in (0, 1)$. Hence there must be a cycle. Since the pre-processing step has eliminated all cycles then no connected component has 0 floating vertices. The result then follows from applying Lemma C.1 to each connected component of $G^1$. □

## D CORRECTNESS OF FINDCCC

PROOF OF LEMMA 7. Consider $\mathcal{P} = \langle s_i \rightsquigarrow t_i \rangle_{i \in [q]}$. Assign an arbitrary color to $s_1$, say $C(s_1) = 1$. Note here $s_1$ and $t_q$ are the two partnerless vertices. Because $C$ only has two colors, this immediately determines the color of $t_1$, which depends on whether $s_1$ and $t_1$ are same-side vertices. Again, this immediately determines the color of $s_2$, which in turn determines the color of $t_2$, and so on. Since $s_1$ and $t_q$ are partnerless, their colors can be whatever they need to be to satisfy the first property. The vertices are colored in the order $s_1, t_1, s_2, t_2, \ldots, s_q, t_q$. Thus, if $\mathcal{P}$ is a CCW this greedy scheme efficiently finds a roundable coloring in time $O(|L| + |R|)$.

Instead suppose $\mathcal{P}$ is a CCC. Once more, the same greedy scheme starting by coloring $s_1$ will satisfy all roundable coloring properties, except maybe for $C(t_q) = C(s_1)$. We now verify this. Observe that the greedy algorithm ensures

$$\forall i \in [q], \ C(t_i) = C(s_i) \cdot (-1)^{\mathbb{1}(t_i \perp s_i)}$$

and

$$\forall i \in [q-1], \ C(s_{i+1}) = C(t_i) \cdot (-1)^{\mathbb{1}(t_i \not\perp s_{i+1})}$$

where $\mathbb{1}(\cdot)$ equals 1 if "$\cdot$" is true and 0 otherwise; this is a slight abuse of notation since $\perp$ is a relation but we are treating it as a boolean function. Expanding by repeated application of the above observations, we have

$$C(t_q) = C(s_1) \cdot (-1)^{\sum_{i \in [q]} \mathbb{1}(t_i \not\perp s_i) + \sum_{i \in [q-1]} \mathbb{1}(t_i \not\perp s_{i+1})}$$
$$= C(s_1) \cdot (-1)^{p + s_{q-1}}$$

where $p$ is the number of same-side paths and $s_{q-1}$ is the number of same-side partners not counting the pair $t_q$ and $s_1$. Then ensuring we have a roundable coloring reduces to ensuring that $t_q \perp s_1 \implies$

$p + s_{q-1}$ is even and $t_q \not\perp s_1 \implies p + s_{q-1}$ is odd. Letting $s$ be the total number of same-side partners, the above is equivalent to asking $p + s$ be even, which we now prove.

Let $d$ be the number of different-side paths, and let $c_L, c_R, c_{LR}$ respectively be the number of left-left, right-right, and left-right partner pairs. So,

$$p + d = q = s + c_{LR}. \tag{15}$$

Let $n_L$ be the number of left vertices in the CCC. Clearly, $n_L = 2c_L + c_{LR}$. Look at $n_L - d$, this is the number of left vertices remaining after removing the different-side paths in $\langle s_i \rightsquigarrow t_i \rangle_{i \in [q]}$. Since these vertices must be covered by same side paths we must have $n_L - d$ even. Then, with all congruences taken modulo 2,

$$0 \equiv n_L - d \equiv 2c_L + c_{LR} - d \equiv c_{LR} - d.$$

Plugging the above into (15) gives $p = s + c_{LR} - d \equiv s$. Therefore, $s + p \equiv s + s \equiv 0$. □

PROOF OF LEMMA 6. For a vertex $b \in L(E^r) \cup R(E^r)$, let $K(b)$ be the set of vertices in the connected component of $G^r$ containing $b$. Let $V_r$ be the *sequence* of vertices $V$ at the beginning of iteration $r$ (i.e., corresponding to $\Psi$ in CCWALK). For a vertex $b$ We use $V_r - b$ to denote $V_r$ without vertex $b$. Like in CCWALK, "$\oplus$" denotes sequence concatenation. Let $a_r$ and $z_r$ be the first and last vertices of $V_r$; note $a_r$ does not change over iterations. We prove the correctness of CCWALK with the aid of the following loop invariants maintained at the beginning of each iteration $r$ of the while-loop.

**(I1)** $z_r$ has no partners in $V_r$.
**(I2)** $\forall b \in V_r - z_r$, $b$ has exactly one partner in $V_r$.
**(I3)** $a_r$ is the only vertex from $V_r$ in $K(a_r)$.
**(I4)** $\forall b \in V_r - a_r$, there are exactly two vertices from $V_r$ contained in $K(b)$.

Proceed by induction. When $r = 1$, $V_r = \langle a \rangle$ so $a_r = a = z_r$ so all invariants are (vacuously) true. Let $P(k)$ be the predicate saying all invariants hold at the beginning of iteration $k \geq 1$. We assume $P(k)$ and show $P(k + 1)$.

If there is an iteration $k+1$, then CCWALK did not return during iteration $k$ and must have added $\langle s_k, t_k \rangle$ to $V_k = \langle t_1, s_2, t_2, \ldots, s_{k-1}, t_{k-1} \rangle$. If $z_{k+1} = t_k$ had a partner in $V_k \oplus \langle s_k \rangle$ then iteration $k$ would have been the last as a CCC would have been returned. Therefore **(I1)** holds at the beginning of iteration $k + 1$.

By $P(k)$, $\forall b \in V_k - t_{k-1}$, $b$ has exactly one partner in $V_k$ and $t_{k-1}$ has zero partners in $V_k$. By construction $s_k$ is selected to be the partner of $t_{k-1}$ so now $t_{k-1}$ and $s_k$ have exactly one partner each in $V_{k+1}$. Therefore **(I2)** holds at the beginning of iteration $k + 1$.

If $V_{k+1}$ were to not meet **(I3)** then it must mean that either $s_k \in K(a_k)$ or $t_k \in K(a_k)$. But in this case the connected component $K(a_k)$ was revisited during iteration $k$ and a CCC was returned.

By construction $K(s_k) = K(t_k)$. Moreover, $\forall b \in V_k$, $K(s_k) \neq K(b)$ otherwise $K(b)$ was revisited and iteration $k$ would have been the last. Therefore, **(I4)** continues to hold.

Moreover note the while-loop runs at most $O(|L| + |R|)$ many times since revisiting a connected component of $G_r$ causes the function to return.

Next we leverage the loop invariants to prove CCWALK returns valid CCC's. First observe that by construction of $V$, Properties 1) and 4) of a CCC are always immediate. CCWALK returns CCC's

when a connected component is revisited or when the added $t_i$ already has partners present in $V$. Fix some iteration $r \geq 1$.

Suppose a connected component $C_k$, $k < r$ is revisited. Then $\langle s'_k \rightsquigarrow t_k, \ldots, s_{r-1} \rightsquigarrow t_{r-1} \rangle$ is returned. By **(I1)**, $t_{r-1}$ and $s'_k$ are each others only partners. Recall by construction $t_k \sim s_{k+1}, t_{k+1} \sim s_{k+1}, \ldots, s_{r-1} \sim t_{r-1}$, so by **(I2)** it follows that each of $t_k, s_{k+1}, t_{k+1}, \ldots, s_{r-1}, t_{r-1}$ has at exactly one partner amongst themselves. Therefore, Property 2) of CCC's holds. It remains to check Property 3). By **(I4)** and the fact that there are paths $s_p \rightsquigarrow t_p$, for $k < p < r$, each $s_p \rightsquigarrow t_p$ belongs to a distinct connected component. $s'_k \rightsquigarrow t_k$ must belong to a unique connected component different from each $s_p \rightsquigarrow t_p$ ; otherwise there was a connected component containing $s_p$, $t_p$, and $s_k$ contradicting one of **(I3)** and **(I4)** (depending on whether $k = 1$ or $k > 1$).

Instead suppose a CCC is returned because $t_r$ had a partner $b \in V_r \oplus (s_r)$. Let $b$ be the last vertex in $V_r \oplus (s_r)$ such that $b \sim t_r$. If $b = s_r$ then $\langle s_r \rightsquigarrow t_r \rangle$ is clearly a CCC. So suppose $b \in V_r$. It must be that $b = s_p$ for some $p < r$; otherwise $b$ is not the last such vertex. So focus on proving $\langle s_p \rightsquigarrow t_p, \ldots, s_r \rightsquigarrow t_r \rangle$ is a CCC. Property 2) follows because $t_{r-1}$ and $s_r$ are each other's only partners by **(I1)**; $t_r$ and $s_p$ are each other's only partners because by **(I2)** $s_p$ previously had only partner $t_{p-1}$ but we have cut it out from the CCC; and the rest of the pairs have unique partners by **(I2)**. Lastly, Property 3) holds because $K(s_r)$ was not a revisited CC so $s_r$ and $t_r$ belong to a distinct CC, and the rest of the path endpoints belong to distinct CC's by **(I4)**.

The last remaining case is that where the two half-CCW's overlap. This can be resolved into a CCC in the manner already described in the main text under the paragraph *Uncrossing the half-CCW's of* Section 5.2.

*Runtime of* FindCCC.. We conclude with comments about the runtime of FindCCC. We can build a hash-map mapping vertices to a set of all their floating partners. This hash-map can be constructed in time $O(|L| + |R|)$. Similarly, we can build a set to keep track of visited connected components. Finding the partner $s_i$ of $t_{i-1}$ can be done in $O(1)$ time by checking the hash-map, and finding the vertex $t_i$ in $K(s_i)$ can be done by starting a depth first search from $t_i$ until a floating vertex is reached. After rounding a the CCC/CCW remove vertices that became settled from their respective sets of floating partners. The depth first searches starting from each $t_i$ altogether visit each vertex and edge at most $O(1)$ times before returning and each vertex is removed from the hash-map's set of floating partners at most once. Therefore, CCWalk runs in time $O(|L| + |R|)$. FindCCC calls CCWalk $O(1)$ times and resolving the two-half CCW's can be thought of as another run of CCWalk, as argued above. Therefore, FindCCC finishes in time $O(|L|+|R|)$. □

# E PROOF OF THEOREM 3

Lemma E.1. *BarterDR satisfies* **(J1)**.

Proof. The property holds trivially for $r = 0$. Recall $r = 1$ corresponds to the output of the pre-processing step. This fact is proved in [15]. Therefore, focus on some fixed $r > 1$ and proceed by induction. Fix $e \in E$ and the CCC/CCW to be rounded $\mathcal{P} = \langle s_i \rightsquigarrow t_i \rangle_{i \in [q]}$. Proceed by considering the following two events.

*Event A:* $e$ does not appear in $\mathcal{P}$ so $e$ does not change this iteration. Thus by the induction hypothesis $\mathbb{E}[x_e^{r+1} \mid (x_e^r = z) \wedge A] = z$.

*Event B:* $e$ appears in $\mathcal{P}$, say, in path $s_i \rightsquigarrow t_i$ for a fixed $i$. Recall values $\alpha$ and $\beta$ from (5) are fixed and $x_e^r$ is modified according to (6). Assuming $e \in M_{-1}^i$ then

$$\mathbb{E}[x_e^{r+1} \mid (x_e^r = z) \wedge B] = z + \frac{\alpha}{v_i}\left(\frac{\beta}{\alpha + \beta}\right) - \frac{\beta}{v_i}\left(\frac{\alpha}{\alpha + \beta}\right) = z.$$

The same holds if instead $e \in M_1^i$. Hence

$$\mathbb{E}[x_e^{r+1} \mid (x_e^r = z)] = \mathbb{E}[x_e^{r+1} \mid (x_e^r = z) \wedge B] \cdot \Pr(B)$$
$$+ \mathbb{E}[x_e^{r+1} \mid (x_e^r = z) \wedge A] \cdot \Pr(A)$$
$$= z(\Pr(A) + \Pr(B)) = z.$$

Let $Z$ be the set of possible values for $x_e^r$.

$$\mathbb{E}[x_e^{r+1}] = \sum_{z \in Z} \mathbb{E}[x_e^{r+1} \mid (x_e^r = z)] \cdot \Pr(x_e^r = z)$$
$$= \sum_{z \in Z} z \cdot \Pr(x_e^r = z) = \mathbb{E}[x_e^r].$$

By the IH, then $\mathbb{E}[x_e^{r+1}] = x_e^0$. □

Lemma E.2. *BarterDR satisfies* **(J2)**.

Proof. The property holds trivially for $r = 0$. Recall $r = 1$ corresponds to the output of the pre-processing step. This fact is proved in [15]. Therefore, focus on some fixed $r > 1$ and proceed by induction. Fix $a \in L \cup R$ and the CCC/CCW to be rounded $\mathcal{P} = \langle s_i \rightsquigarrow t_i \rangle_{i \in [q]}$. Recall $V(\mathcal{P})$ denotes the endpoints of the paths of $\mathcal{P}$. Proceed by cases.

*Case A:* $a \notin V(\mathcal{P})$. Then either $a$ does not appear in $\mathcal{P}$ or $a$ appears in $\mathcal{P}$ but with two edges incident on it. In the former case clearly $x_a^{r+1} = x_a^r$. In the latter case, the change of each incident edge is equal in magnitude and opposite in sign (since one edge belongs to $M_{-1}^i$ and the other to $M_1^i$) therefore $x_a^{r+1} = x_a^r$ as well. Thus by the IH $\lfloor x^0(a) \rfloor \leq x^{r+1}(a) \leq \lceil x^0(a) \rceil$.

*Case B:* $a \in V(\mathcal{P})$. There is a single incident edge $e \in N(a)$. Without loss of generality, said edge belongs path $s_i \rightsquigarrow t_i$ and thus to $M_{-1}^i$ (the proof for $M_1^i$ is identical). Then either $x^{r+1}(a) = x^r(a) + \alpha/v_i$ or $x^{r+1}(a) = x^r(a) - \beta/v_i$. In either case, by definition of $\alpha$ and $\beta$ (i.e., (5)), $\alpha$ and $\beta$ are small enough that $\lfloor x^r(a) \rfloor \leq x^{r+1}(a) \leq \lceil x^r(a) \rceil$. Observe $\lfloor x^0(a) \rfloor = \lfloor x^r(a) \rfloor$ and $\lceil x^0(a) \rceil = \lceil x^r(a) \rceil$.

Having handled exhaustive cases, the proof is complete. □

Lemma E.3. *BarterDR satisfies* **(J3)**.

Proof. The property holds trivially for $r = 0$. Recall $r = 1$ corresponds to the output of the pre-processing step. This fact is proved in [15]. Therefore, focus on some fixed $r > 1$ and proceed by induction. Fix a vertex $a$ and a subset of edges $S$ incident on $a$ like in **(J3)**. Also fix the CCC/CCW to be rounded $\mathcal{P} = \langle s_i \rightsquigarrow t_i \rangle_{i \in [q]}$. Proceed based on the following events.

*Event A:* no edge in $S$ has its value modified. Then $\mathbb{E}[\prod_{e \in S} x_e^{r+1} \mid A] = \mathbb{E}[\prod_{e \in S} x_e^r \mid A]$.

*Event B:* two edges $e_1, e_2 \in S$ have their values modified. Said edges must both belong to $s_i \rightsquigarrow t_i$, for some fixed $i$, with one

belonging to $M^i_{-1}$ and the other to $M^i_1$; say $e_1 \in M^i_1$ and $e_2 \in M^i_{-1}$. Then

$$(x^{r+1}_{e_1}, x^{r+1}_{e_2}) = \begin{cases} (x^r_{e_1} + \alpha/v_i, x^r_{e_2} - \alpha/v_i) & \text{with probability } \beta/(\alpha + \beta) \\ (x^r_{e_1} - \beta/v_i, x^r_{e_2} + \beta/v_i) & \text{with probability } \alpha/(\alpha + \beta) \end{cases}$$

where $\alpha$ and $\beta$ are fixed per (5). Let $S_1 = S - \{e_1, e_2\}$. Then

$$\mathbb{E}\left[\prod_{e \in S} x^{r+1}_e \mid (\forall e \in S, x^r_e = z_e) \wedge B\right]$$

$$= \mathbb{E}\left[x^r_{e_1} \cdot x^r_{e_2} \mid (\forall e \in S, x^r_e = z_e) \wedge B\right] \prod_{e \in S_1} z_e.$$

The above expectation can be written as $(\Psi + \Phi) \prod_{e \in S_1} z_e$, where

$$\Psi = (\beta/(\alpha + \beta)) \cdot (z_{e_1} + \alpha) \cdot (z_{e_2} - \alpha) \text{ and}$$
$$\Phi = (\alpha/(\alpha + \beta)) \cdot (z_{e_1} - \beta) \cdot (z_{e_2} + \beta).$$

It is easy to s how $\Psi + \Phi \leq z_{e_1} z_{e_2}$. Thus, for any fixed $\{e_1, e_2\} \subseteq S$ and for any fixed $(\alpha, \beta)$, and for fixed values of $z_e$, the following holds:

$$\mathbb{E}\left[\prod_{e \in S} x^{r+1}_e \mid (\forall e \in S, x^r_e = z_e) \wedge B\right] \leq \prod_{e \in S} z_e.$$

Hence, $\mathbb{E}[\prod_{e \in S} x^{r+1}_e \mid B] \leq \mathbb{E}[\prod_{e \in S} x^r_e \mid B]$.

*Event C:* exactly one edge in the set $S$ has its value modified. Let $C$ denote the event that edge $e_1 \in S$ has its value changed in the following probabilistic way

$$x^{r+1}_{e_1} = \begin{cases} x^r_{e_1} + \alpha & \text{with probability } \beta/(\alpha + \beta) \\ x^r_{e_1} - \beta & \text{with probability } \alpha/(\alpha + \beta). \end{cases}$$

Thus, $\mathbb{E}[x^{r+1}_{e_1} \mid (\forall e \in S, x^r_e = z_e) \wedge C] = z_{e_1}$. Letting $S_1 = S - \{e_1\}$, we get that $\mathbb{E}[\prod_{e \in S} x^{r+1}_e \mid (\forall e \in S, x^r_e = z_f) \wedge C]$ equals

$$E[x^{r+1}_{e_1} \mid (\forall e \in S, x^r_e = z_f) \wedge C] \prod_{e \in S_1} z_e = \prod_{e \in S} z_e.$$

Since the equation holds for any $e_1 \in S$, for any values of $z_e$, and for any $(\alpha, \beta)$, we have $\mathbb{E}[\prod_{e \in S} x^{r+1}_e \mid C] = \mathbb{E}[\prod_{e \in S} x^r_e]$. □

PROOF OF THEOREM 3. By Lemmas E.1 and E.2 BARTERDR satisfies **(P1)** and **(P2)**. Let $T$ be the last iteration of BARTERDR. From Lemma E.3 we have

$$\Pr(\bigwedge_{e \in S}(X_e = 1)) = \mathbb{E}[\prod_{e \in S} x^{T+1}_e] \leq \mathbb{E}[\prod_{e \in S} x^1_e] = \prod_{e \in S} x^0_e = \prod_{e \in S} \Pr(X_e = 1).$$

The proof for $c = 0$ (i.e., $\Pr(X_e = 0)$) is identical. Therefore, BARTERDR satisfies **(P3)**. □

# F PROOF OF THEOREM 1

PROOF OF LEMMA 8. Given a BarterSV instance, let $\text{OPT}_{\text{IP}}$ and $\text{OPT}_{\text{LP}}$ be the optimal objectives of the corresponding IP (1) and the corresponding BarterSV-LP. Let $\{X^*_e\}_{e \in E}$ and $\{x^*_e\}_{e \in E}$ be optimal solutions to the IP and LP, respectively. Then $\text{OPT}_{\text{IP}} = \sum_{e \in E} w_e X^*_e \leq \sum_{e \in E} w_e x^*_e = \text{OPT}_{\text{LP}}$. Per Theorem 3, BARTERDR satisfies **(P1)** when rounding $\{x^*_e\}_{e \in E}$ to $\{X_e\} \in \{0, 1\}^{|E|}$. Therefore, $\mathbb{E}[\sum_{e \in E} w_e X_e] = \sum_{e \in E} w_e \mathbb{E}[X_e] = \sum_{e \in E} w_e x^*_e = \text{OPT}_{\text{LP}}$.

By the linearity of expectation and **(P1)**,

$$\mathbb{E}[D_i] = \mathbb{E}[\sum_{a \in L_i} X(a) v_a - \sum_{b \in R_i} X(b) v_b]$$

$$= \sum_{b \in L_i} \mathbb{E}[X(b)] v_b - \sum_{a \in R_i} \mathbb{E}[X(a)] v_a$$

$$= \sum_{b \in L_i} x^*(b) v_b - \sum_{a \in R_i} x^*(a) v_a = D^0_i = 0.$$

The last equation follows because $\{x^*_e\}_{e \in E}$ satisfies (1d) per Lemma E.1. □

PROOF OF LEMMA 9. If no vertex from $L_i \cup R_i$ appears in the CCC/CCW's endpoints $V(\mathcal{P}) := \bigcup_{i \in [q]}\{s_i, t_i\}$ then we are done. So suppose $a \in L_i \cup R_i$ and $a \in V(\mathcal{P})$ on this $r$-th rounding iteration. By assumption there exists another floating $b' \in L_i \cup R_i$ in iteration $r$ when $\langle s_i \rightsquigarrow t_i \rangle_{i \in [q]}$ was constructed. Therefore, $a$ is not partnerless hence it cannot be the endpoint of a CCW so there exists $b \in V(\mathcal{P})$ such that $a \sim b$. Moreover, by property 2 of the definition of a CCC/CCW, said $b$ is unique. Therefore, $a$ and $b$ are the *only* vertices in $V(\mathcal{P})$ affecting $D_i$ this iteration $r$; i.e., $\forall d \in (L_i \cup R_i) - \{a, b\}$, $x^r(d) = x^{r+1}(d)$. Since $a \sim b$, we may assume without loss of generality that $a = t_k$ and $b = s_{k+1}$ for some $k \in [q]$ (or $k \in [q - 1]$ for a CCW); recall $s_{q+1} \equiv s_1$. We know $C(a) = C(b)$ if and only if $a$ and $b$ belong to opposite graph sides where $C$ is the valid coloring function corresponding to $\mathcal{P}$, which can be fixed efficiently per Lemma 7.

Consider the two possible rounding events, described in (6). Call these events $\theta_1$ and $\theta_2$. Suppose $a$ and $b$ are opposite-side vertices, hence $C(a) = C(b)$. Focus on event $\theta_1$ (the proof for $\theta_2$ is exactly the same but replacing $\alpha$ with $-\beta$). Under event $\theta_1$ we have

$$x^{r+1}(a) = x^r(a) - C(a)\frac{1}{v_a}\alpha \quad \text{and} \quad x^{r+1}(b) = x^r(b) - C(b)\frac{1}{v_b}\alpha.$$

Note the factor of "$-C(a)$" appears because $a$ belongs to $M^i_{C(a)}$ for some $i$. Conveniently, this leaves us with

$$x^{r+1}(a) v_a - C(a)C(b) x^{r+1}(b) v_b$$

$$= x^r(a) v_a - C(a)C(b) x^r(b) v_b - C(a)\alpha + C(a)\alpha \quad (16)$$

$$= x^r(a) v_a - C(a)C(b) x^r(b) v_b, \quad (17)$$

using the fact $C(b) \cdot C(b) = 1$. Without loss of generality let $a \in L_i$. Therefore, expanding $D^{r+1}_i$:

$$D^{r+1}_i = \sum_{s \in L_i} x^{r+1}(s) v_t - \sum_{t \in R_i} x^{r+1}(t) v_t. \quad (18)$$

Having fixed $a \in L_i$, we know $C(a)C(b) = 1$ if and only if $b \in R_i$. Thus, take out $x^{r+1}(a)$ and $x^{r+1}(b)$ from the sums and substitute (17) to have

$$\sum_{s \in L_i - \{a,b\}} x^{r+1}(s) v_t - \sum_{t \in R_i - \{b\}} x^{r+1}(t) v_t + x^r(a) v_a - C(a)C(b) x^r(b) v_b.$$
$$(19)$$

Now observe that $x^{r+1}(p) = x^r(p)$ for all $p \in (L_i \cup R_i) - \{a, b\}$. Moreover, $b \in R_i$ if and only if $C(a)C(b) = 1$, so we can reabsorb the terms "$x^r(a) v_a$" and "$x^r(b) v_b$" into their respective summations; thus yielding $\sum_{s \in L_i} x^r(s) v_t - \sum_{t \in R_i} x^r(t) v_t$. But this is precisely $D^r_i$, which we've assumed to be 0. □

# G HARDNESS OF BarterSV

We first prove Theorem 2: it is NP-Hard to find any non-empty allocation satisfying $D_k = 0$ for all agents $k$. By non-empty we mean the corresponding LP solution $x \neq 0$, i.e., at least one agent gives away an item. The proof proceeds by reducing from the NP-hard problem of PARTITION.

**Definition 4** (PARTITION). A PARTITION instance takes a set $S = \{a_1, a_2, \ldots, a_n\}$ of $n$ positive integers summing to an integer $2T$. The goal of PARTITION is to determine if $S$ can be partitioned into disjoint subsets $S_1$ and $S_2$ such that each subset sums exactly to an integer $T$.

LEMMA G.1. *Given a* PARTITION *instance, it can be reduced in polynomial time to a corresponding* BarterSV *instance with two agents.*

PROOF. Consider an instance $I = (S, 2T)$ of partition problem where $S = \{a_1, a_2, \ldots, a_n\}$ such that $\sum_{i \in [n]} a_i = 2T$ and $a_i$ is an integer, for all $i \in [n]$. Given an instance $I$ of PARTITION, the BarterSV instance constructed is as follows.

Let the set of items $\mathcal{I} = \{i_1, i_2, \ldots, i_n, i_{n+1}\}$ with item values $v_j := a_j$ for each item $i_j, j \in [n]$ and $v_{n+1} := T$ for item $i_{n+1}$. There are two agents $A = \{1, 2\}$, where agent 1 has item lists $H_1 := \{i_1, \ldots, i_n\}$ and $W_1 := \{i_{n+1}\}$. Symmetrically, agent 2 has item lists $W_2 := \{i_j : j \in [n]\}$ and $H_2 := \{i_{n+1}\}$. The particular weights of allocating items (i.e., $w_e$ in the bipartite graph) do not matter as we only care about whether some non-empty allocation exists. □

Recall the goal is to show there exists a *non-empty* allocation such that for each agent $k \in [2]$, $D_k = 0$ if and only if the corresponding PARTITION instance has a solution.

LEMMA G.2. *There exists a polynomial time algorithm to find a non-empty allocation of items with $D_k = 0$ for each agent $k$ in the* BarterSV *instance if and only if there exists a polynomial time algorithm to the corresponding* PARTITION *instance.*

PROOF. **Forward direction** (PARTITION $\implies$ BarterSV). Given a solution $(S_1, S_2)$ to the PARTITION instance, the corresponding BarterSV instance has a solution in the following manner. Allocate the items $\{i_j : j \in S_1\}$ in the have-list of agent 1 to agent 2 and allocate the item $i_{n+1}$ to agent 1. Thus, the value of the items received and given by both the agents is exactly $T$ resulting in a non-empty allocation with $D_1 = D_2 = 0$.

**Backward Direction** (BarterSV $\implies$ PARTITION). Take a *non-empty* allocation of items $\mathcal{I}$ to each agent $k \in [2]$ with $D_k = 0$. Such a non-empty allocation must have agent 1 giving away their only item, which has value $T$. Therefore $D_1 = 0$ implies agent 1 received $T$ units of value. Let the items agent 1 received be $i_{j_1}, i_{j_2}, \ldots, i_{j_\ell}$, letting $J = \{j_1, \ldots, j_\ell\}$. Thus, $\sum_{p \in J} v_p = T$. Therefore, the corresponding partition instance has solution $S_1 = \{a_p : p \in J\}$ and $S_2 = \{a_p : p \notin J\}$. □

Thus, Lemmas G.1 and G.2 show that it is NP-hard to find a non-empty allocation of items with $D_k = 0$ for any agent $k$. That is, Theorem 2.

## G.1 Additional hardness results

The prior hardness result showed that finding *any* non-empty allocation is NP-hard but not necessarily *strongly* NP-hard since PARTITION is weakly NP-hard. We can additionally show the decision version of BarterSV (i.e., does there exist a non-empty allocation with utility $\geq K$) is strongly NP-hard.

THEOREM 4. *The decision version of* BarterSV *is strongly NP-Hard.*

**Definition 5** (3-PARTITION). Given a set $S = \{a_1, a_2, \ldots, a_{3m}\}$ of $3m$ positive integers summing to an integer $mT$ and $a_i \in (T/4, T/2)$. The goal of 3-PARTITION is to determine if $S$ can be partitioned into $m$ disjoint subsets $S_1, S_2, \ldots, S_m$, such that each subset sums exactly to an integer $T$.

Note that each subset $S_i, i \in [m]$ in the partition can have exactly 3 integers each.

LEMMA G.3. *Given any 3-PARTITION instance, it can be reduced to a corresponding* BarterSV *instance with $m + 1$ agents in polynomial time.*

PROOF. Consider an instance $I = (S, T)$ of 3-PARTITION where $S = \{a_1, a_2, \ldots, a_{3m}\}$ such that $\sum_{i \in [3m]} a_i = mT$ and $a_i$ is an integer, for all $i \in [3m]$. Given an instance $I$ of 3-PARTITION we construct an instance of BarterSV as follows:
Suppose that the set of items $\mathcal{I} = \{i_1, i_2, \ldots, i_{3m}, i_{3m+1}, \ldots, i_{4m}\}$ where the values are $v_j := a_j$ for each item $i_j, j \in [3m]$ and $v_{3m+k} := T$ for each item $i_{3m+k}, k \in [m]$. There are a set of $m + 1$ agents $A = \{1, 2, \ldots, m + 1\}$, where each agent $k \in [m]$ consists of the items $\{i_{3m+k}\}$ and $\{i_1, i_2, \ldots, i_{3m}\}$ in its have-list and wish-list respectively. The agent $m + 1$ has $m$ items in its wish-list $\{i_{3m+k} : k \in [m]\}$ and $\{i_1, i_2, \ldots, i_{3m}\}$ in its have-list. Let the weight of allocating item $i_j$ from agent $a$ to any valid agent $b$ is given by $v_j$. Therefore, the goal of the *decision version* of BarterSV problem is to determine if there exists a re-allocation of items such that for each agent $k \in [m + 1]$, $D_k = 0$ and the total *allocation utility* is $(2m)T$. □

The following lemma is crucial in establishing the hardness of BarterSV.

LEMMA G.4. *There exists a polynomial time solution to* BarterSV *problem instance iff there exists a polynomial time solution to the corresponding instance of the 3-PARTITION.*

PROOF. **Forward Direction ( 3-PARTITION $\implies$ BarterSV)** Given a solution $S_1, S_2, \ldots, S_m$ to 3-PARTITION, we can find a solution to the corresponding instance of BarterSV in the following manner. We allocate, the items $\{i_j : j \in S_k\}$ in the have-list of agent $m + 1$ to agent $k$ for each $k \in [m]$ and allocate all the items $\{i_{3m+k} : k \in [m]\}$ to agent $m + 1$. Thus, the value of the items received and given by agent $m+1$ is exactly $mT$ resulting in $D_{m+1} = 0$. Further the value of items assigned to each agent $k \in [m]$ is exactly $T$ which is equal to the value of item given. Observe that for each agent $k \in [m]$, the total value of items received and given are exactly $T$, therefore $D_k = 0$. Therefore, the *utility* of the allocation is exactly $mT + mT = 2mT$ and $D_k = 0$ for each agent $k \in [m + 1]$.

**Backward Direction** (BarterSV $\implies$ **3-PARTITION**) Given a non-empty allocation of items $\mathcal{I}$ to each agent $k \in [m + 1]$

with $D_k = 0$ such that the total value of items reallocated is $2mT$. Then each agent $k \in [m]$ has respectively received items from the partition $H_{m+1,1}, H_{m+1,2}, \ldots, H_{m+1,m}$ of the have-list of agent $m + 1$ such that $\sum_{j \in H_{m+1,k}} v_j = T$ for each $k \in [m]$. Recall that the items $H_{m+1} = \{i_1, i_2, \ldots, i_{3m}\} = \dot{\bigcup}_{k \in [m]} H_{m+1,k}$. Therefore, we can create a solution to the corresponding 3-Partition instance

by choosing $S_k = \{a_j : j \in H_{m+1,k}\}$ for each $k \in [m]$. Notice that $\sum_{k \in [m]} |S_k| = 3m$, $a_j \in (T/4, T/2)$ implies $|S_k| = 3$ for all $k$, and for all $k \in [m]$, $\sum_{j \in S_k} a_j = T$. Therefore, $S_1, S_2, \ldots, S_m$ is both a partition of $S$ and a valid solution to the problem. □

Thus Lemma G.4 shows the decision version of the BarterSV is NP-hard. Therefore, Theorem 4 follows.

