# OpenReview forum: "Barter Exchange with Shared Item Valuations"
_ACM.org/TheWebConf/2024/Conference — TheWebConf24_

### Official Review · Reviewer_r9WQ · 2023-11-17

**Novelty:** 4
**Technical Quality:** 4

**Review:**

This paper introduces and studies a centralized barter exchange problem (BarterSV) with a set of agents and items in which each item has a subjective positive value agreed upon by all agents. The goal is to maximize the total value of the allocation, under the constraint that each agent’s total received value exactly equals their total given value. This paper further reduces this problem to a value-balanced matching, which can be written as an Integer Program. By relaxing the Integer Program to a natural Linear Program, the solution of LP can be considered as an (optimal) fractional allocation of BarterSV. This paper further proposed a dependent rounding (built on GKPS-DR) that guarantees good ex-post allocations (The difference between each agent’s total given value and total received value can be bounded by the most valuable item).

Strengths:
1)	This paper introduces the BarterSV problem, which is natural and has many applications in the real world, e.g., GAMESWAP and BARDAGAMEEXCHANGE.
2)	This paper shows multiple connections between the BarterSV problem with the well-studied bipartite matching problem: 1) it formulates the problem as a bipartite-matching problem with barter constraints; 2) The proposed algorithm BARTERDR is based on LP rounding, which is a well-developed technique in the bipartite matching.
3)	The proposed algorithm BARTERDR guarantees good ex-post properties, which is not guaranteed in the previous algorithms like GKPS-DR.

Weaknesses:
1）	The novelty of the article seems limited. Although it proposes the BarterSV problem and studies it relatively completely, most of the techniques it used were based on existing work.
2）	The structure of this paper can be improved. As far as I know, the WebConf requires that the main paper should not exceed 8 pages. However, the main body of this paper is only a bit over six pages. At the same time, many relevant proofs are placed in the appendix. This might not help readers to get a full understanding of this paper. I would suggest the authors consider providing some important proofs (or providing a more intuitive proof sketch) in the main body.
3）	To me, Section 6 is redundant. The entire section can be regarded as a corollary, instead of providing more interesting results. It also did not provide a more detailed discussion.

**Questions:**

1) Have you tried more general settings, such as agents may have different valuation functions? Is it possible to extend the proposed algorithms or techniques to these more general settings?

2) Since you have mentioned fairness, are there any possible results regarding fairness? What directions regarding fairness are interesting to explore?

**Reviewer Confidence:**

3: The reviewer is confident but not certain that the evaluation is correct

**Scope:**

3: The work is somewhat relevant to the Web and to the track, and is of narrow interest to a sub-community

---

### Official Review · Reviewer_4PbT · 2023-11-23

**Novelty:** 4
**Technical Quality:** 3

**Review:**

## Summary:

The paper works on a barter exchange problem, where agents seek to swap items for other items in their wishlist without using payments to maximize the agents' collective utility. The paper introduces a specific barter exchange problem where the utility is maximized, and the total value of each agent does not change before and after the reallocation. They find that this problem is NP-Hard, and most of the paper shows how to compute allocations efficiently, in which each agent loses a value of at most one item.

## Evaluation

### Pros:

1. The paper is working on an interesting problem, with a significant connection to the web conference.

### Cons

1. The paper is messily written and structured in a way that compromises its message. Even the analysis of Theorem 1, the technically most important part of the paper, is messily presented, without a sketch of proof to guide the reader and very little intuition. Importantly, I don't think the main message of the paper is given in a way that is ready for publication. Also, the problem is not properly introduced.
2. While being non-trivial, the proposed results look relatively weak. The paper proposes one approximation algorithm, which is better than an algorithm from the literature designed for a different problem, and this is the only comparison given, which is relatively weak. It would be much more robust if we knew some bounds on the approximability of the problem, i.e., what is the minimum net value loss we must suffer if we want a polynomial time algorithm?

Some detailed comments:

- line 94: "a subset" repeated words.

- lines 144-145: Why do we need an allocation to be invalid when an agent receives more copies than desired? While this is an admirable goal for an algorithm, hardwiring this on the allocation's validity might lead us to miss some helpful allocation (maybe write an example here!).

- Line 140: what is the original problem? In general, the problem proposed in a way that looks new, but it is also seems that it has connections to the literature which are not presented (or partially presented) in the paper.

- Definition 1: It is unclear what $L_i$ and $R_i$ denote in the definition! This could be crucial!! I guess that this is described in lines 172 and 173, but it should be clearly noted in the definition!

- Line 243 (degree preservation property): $X(a)$ and $x(a)$ is not defined up this point! In general I believe the GKPS-DR procedure and its connection to the main problem should be discussed further.

- Line 352: probably partners should have been emphasized.

- Definition 2 is vital for the paper, yet very few intuition is given to the reader around it.

- Line 871: what is U and V?

- The setting for the proofs of Lemmas 1 and 2 is somewhat problematic. More specifically, part of what is proven is used as granted in the proofs! I don't believe the proofs are wrong, yet they are not written in a fault-proof way! This compromises the quality of the paper.

- The references to the Appendix are very vague. Given the size of the appendix, the authors should have tried to be more precise when referring the reader.

**Questions:**

1 What is the computational complexity of BarterDR (in line 368, there are some hints about this, but it would be nice to have a formal result on this!)?

**Ethics Review Description:**

-

**Reviewer Confidence:**

2: The reviewer is willing to defend the evaluation, but it is likely that the reviewer did not understand parts of the paper

**Scope:**

4: The work is relevant to the Web and to the track, and is of broad interest to the community

---

### Official Review · Reviewer_Wx3g · 2023-11-23

**Novelty:** 4
**Technical Quality:** 4

**Review:**

The paper introduces and studies BarterSV, a centralized barter exchange problem. In this problem, each item has a value agreed upon by the participating agents. The goal is to find an allocation/exchange that maximizes the collective utility of the allocation such that the total value of each agent’s items before and after the exchange is equal. The paper shows that solving this problem exactly is NP-hard (its decision version is strongly NP-complete) by reducing from the Set Partition Problem (3-Partition Problem). Moreover, the authors provide an efficient randomized rounding algorithm to find an allocation with the same expected objective value of the optimal solution while violating the balance constraints by the value of one item (namely, the value of the most owned or wished-for item). The paper is globally well-written, and the results seem technically correct. There are also typos throughout the paper (e.g., the sentence above Lemma 4), though they do not seem to impact the correctness of the results. Overall, the main contribution of the paper is to show that the dependent randomized rounding of [15] also works well for the BarterSV problem with some additional cycle/path finding subroutines. However, the originality and main novelty of this work, both in the formulation side and the techniques, compared to the past literature is obscure and needs further clarification.

**Questions:**

Q1) What is the main advantage of the current formulation compared to the existing BarterSV models proposed in the past literature, e.g., [2]?

Q2) Often in the BarterSV, the agents are selfish identities, and the exchanges occur in a distributed manner. In that case, what is the practicality of solving this problem in a centralized fashion as done in this work?

Q3) One of the inherent properties of the proposed rounding scheme is its negative correlation property (property 3). It is not clear where this property is used or if it is important for the purpose of the current paper. It is mentioned that the negative correlation property provides some concentration results. However, I did not see any analytical results or discussion to establish such concentration bounds.

Q4) What is the main novelty of the proof technique compared to [15]? For instance, even the choice of parameters \alpha and \beta mimics the same structure as those proposed in [15]. Is the main innovation of the paper on developing subroutines for finding CCC or CCW and performing the rounding over such paths/cycles? Please elaborate.

Q5) Some of the reductions seem to resemble other works [2] (e.g., moving from BarterSV to bipartite matchings in Definition 1). The author should clarify the extent to which these ideas are unique to their work or otherwise cite relevant references.

Q6) Some of the equations and randomized rounding procedures are not well explained (e.g., equations 3 and 4). Perhaps adding one small numerical example and illustrating the rounding process for one state would help.

**Reviewer Confidence:**

3: The reviewer is confident but not certain that the evaluation is correct

**Scope:**

4: The work is relevant to the Web and to the track, and is of broad interest to the community

---

### Official Review · Reviewer_UHdW · 2023-11-24

**Novelty:** 5
**Technical Quality:** 5

**Review:**

The paper studies the barter exchange problem, where each agent wants to swap her own items for items in her wishlist. The price of each item is public. The problem, coined as BarterSV, aims at designing a valid reallocation algorithm for maximizing the total value of the swapped items subject to no agent giving away more value than they received (the barter constraint).

Since BarterSV is NP-Hard (shown in the appendix), the authors consider violating the barter condition, meanwhile expecting the net value loss of each agent will not too large. They design a randomized algorithm with optimal total swapped value in expectation. Moreover, for each agent the net value loss is at most the price of the most expensive item in her havelist and wishlist.

They first formulate BarterSV as an Integer Programming on a bipartite graph, where the decision variables refer to all the edges. A randomized integer solution is obtained by rounding from the corresponding linear programming with probabilities. The previous dependent rounding [J. ACM 06] gave the intuition for the rounding technique, while it can make the net value loss very large. In the paper, the authors adopt dependent rounding in pre-processing to remove cycles in the graph. Then they construct virtue cycles by finding a sequence of paths on distinct connected components in the graph and connecting them by some virtue edges. Finally, they present a roundable coloring on these extreme points of the above paths. The randomized algorithm is achieved by carefully rounding on these fractional values on the edges iteratively.

The model proposed in the paper is interesting. And the technique is also challenging. The idea for rounding is likely be extended to other problems.

Below are some minor comments.
1. Lemma 2. Here the reviewer thinks BarterSV is a special case of VBM but not equivalence.
2. IP(1). It is better giving the notations before the IP, which makes the paper easier to follow.
3. IP(1). The decision variables in the context is x_e, but here is y_e. It is better making them consistent.
4. Line 194, in the context the notation of the bipartite graph is L and R, but here changes to U and V.
5.  Line 347. Typo (The definition of x^r(a) ): e\in E --> e \in N(a).
6. Line 350, line 408 and line 537:  The notation of C(\cdot) is confusing. If the reviewer understood correctly, the first one is the collection of the items; the second one is the connected component and the last one is the color.
7. Line 408 and 409. If the reviewer understood correctly, (2) and (3) are equivalent.

**Questions:**

none

**Reviewer Confidence:**

3: The reviewer is confident but not certain that the evaluation is correct

**Scope:**

4: The work is relevant to the Web and to the track, and is of broad interest to the community

---

### Official Review · Reviewer_BhTG · 2023-11-27

**Novelty:** 6
**Technical Quality:** 5

**Review:**

The authors discuss the problem of barter exchange in this paper. In the problem, each agent enters with a set of items to be offered and a wish list of goods. The goal for agents is to exchange their own items for goods on their wish lists, without the involvement of money provided that the value of goods they offered is comparable to the goods they received. This scenario is common in areas such as social media or online game environments.

The authors formulate this barter exchange problems in the model of Value-Balanced Matching and its corresponding integer programming problem. The first important outcome is that such a barter exchange problem is NP-Hard, so it would be of significance to find some approximate solution. The authors take the approach of first solving the relaxed Linear Programming problem, and then applying a fine-tuned rounding scheme named BarterDR to the linear solution to achieve a favored allocation. The solution returned by the BarterDR satisfies several desirable properties, as the objective value is ex-ante optimal, and each agent will not suffer from any value off in expectation. The solution also has strong ex-post guarantee where the potential value loss for each agent is no more than the price of her most valued item, either in her inventory or in her wish list.

The setting of the problem is novel and interesting, and the algorithm possess customizability because the objective value of the IP can be set arbitrarily. In the paper, the objective is the sum of the value of items involved in the barter exchange, while it can also be set as the number of items exchanged or any other benchmark according to the purpose of the application. Also, there is still more to explore in the setting, as the paper assumes that each item has a universally agreed value. The model can be generalized where each agent may have different valuation on one items, and the objective can be set to the increase of total value of agents after the batter exchange, which has large potential of exploration. Overall, the problem seems exciting, and there is still room for investigation and generalization.

**Questions:**

The following are several minor questions/suggestions I have in reading the paper.

1. In the introduction, near line 93, there seem to be repeated phrases (“the user’s goal is to swap a subset a subset of…”).

2. In the last paragraph of the introduction, near line 122, it states that “…in a worst-case scenario, agents give away all their items and receive none in exchange.” I understand this after reading the paper as that, some – instead of each -- agent will not receive any items in exchange. However, I couldn’t know which case is true before reading the rest of the paper. Perhaps it would be better to clarify this in the intro.

3. In the problem, is it necessary to assume that each H_i and W_i are disjoint? If not, then it would be legal for an agent to exchange items with herself, which adds to the collective utility despite effectively doing nothing.

**Reviewer Confidence:**

2: The reviewer is willing to defend the evaluation, but it is likely that the reviewer did not understand parts of the paper

**Scope:**

3: The work is somewhat relevant to the Web and to the track, and is of narrow interest to a sub-community

---

### Decision · Program_Chairs · 2024-01-22

**Decision:**

Accept

**Comment:**

Most of the reviewers mentioned that problem is interesting, and the algorithm is novel. Moreover, the proposed problem may have several interesting extensions that worth investigating.